# Localized spin-orbit polaron in magnetic Weyl semimetal Co$_3$Sn$_2$S$_2$

Yuqing Xing [1,2,3,8], Jianlei Shen[1,2,8], Hui Chen [1,2,3,8], Li Huang[1,2,3,8], Yuxiang Gao[1,2,3], Qi Zheng[1,2,3], Yu-Yang Zhang [2,3], Geng Li [1,2,3], Bin Hu[1,2,3], Guojian Qian[1,2,3], Lu Cao [1,2,3], Xianli Zhang[1,2,3], Peng Fan[1,2,3], Ruisong Ma[1,2,3], Qi Wang[4], Qiangwei Yin[4], Hechang Lei [4], Wei Ji [4], Shixuan Du[1,2,3,5], Haitao Yang[1,2,3], Wenhong Wang[1,2,5], Chengmin Shen[1,2,3], Xiao Lin[2,1,3], Enke Liu [1,2,5✉], Baogen Shen[1,2,6], Ziqiang Wang[7✉] & Hong-Jun Gao [1,2,3,5✉]

The kagome lattice Co$_3$Sn$_2$S$_2$ exhibits the quintessential topological phenomena of a magnetic Weyl semimetal such as the chiral anomaly and Fermi-arc surface states. Probing its magnetic properties is crucial for understanding this correlated topological state. Here, using spin-polarized scanning tunneling microscopy/spectroscopy (STM/S) and non-contact atomic force microscopy (nc-AFM) combined with first-principle calculations, we report the discovery of localized spin-orbit polarons (SOPs) with three-fold rotation symmetry nucleated around single S-vacancies in Co$_3$Sn$_2$S$_2$. The SOPs carry a magnetic moment and a large diamagnetic orbital magnetization of a possible topological origin associated relating to the diamagnetic circulating current around the S-vacancy. Appreciable magneto-elastic coupling of the SOP is detected by nc-AFM and STM. Our findings suggest that the SOPs can enhance magnetism and more robust time-reversal-symmetry-breaking topological phenomena. Controlled engineering of the SOPs may pave the way toward practical applications in functional quantum devices.

---

[1] Beijing National Center for Condensed Matter Physics and Institute of Physics, Chinese Academy of Sciences, Beijing 100190, People's Republic of China. [2] School of Physical Sciences, University of Chinese Academy of Sciences, Beijing 100190, People's Republic of China. [3] CAS Center for Excellence in Topological Quantum Computation, University of Chinese Academy of Sciences, Beijing 100190, People's Republic of China. [4] Beijing Key Laboratory of Optoelectronic Functional Materials & Micro-Nano Devices, Department of Physics, Renmin University of China, Beijing 100872, People's Republic of China. [5] Songshan Lake Materials Laboratory, Dongguan, Guangdong 523808, People's Republic of China. [6] Institute of Rare Earths, Chinese Academy of Sciences, Jiangxi 341000, China. [7] Department of Physics, Boston College, Chestnut Hill, MA, USA. [8] These authors contributed equally: Yuqing Xing, Jianlei Shen, Hui Chen, Li Huang. ✉email: ekliu@iphy.ac.cn; wangzi@bc.edu; hjgao@iphy.ac.cn

The transition metal based kagome lattice compounds have merged recently as a novel materials platform for unveiling and exploring the rich and unusual physics of geometric frustration, correlation and magnetism, and the topological behaviors of the quantum electronic states[1–18]. These are layered crystalline materials where the transition metal elements occupy the vertices of the two-dimensional network of corner-sharing triangles, supporting electronic band structures with Dirac crossings and nearly flat bands with strong spin–orbit coupling[19–21]. The prototype materials include $Fe_3Sn_2$[1–3], FeSn[4], $Co_3Sn_2S_2$[5,6], CoSn[7], $Mn_3Sn$[8–11], and rare earth (Re) $ReMn_6Sn_6$[12,15]. They exhibit different magnetic ground states, such as ferromagnetic ($Fe_3Sn_2$, $Co_3Sn_2S_2$), antiferromagnetic (FeSn, $Mn_3Sn$), and paramagnetic (CoSn), and often anomalous transport properties of underlying topological origins[5,6,16]. Remarkable phenomena have been reported recently, such as the giant spin-orbit tunability of the Dirac mass and electronic nematicity in $Fe_3Sn_2$[1,2], the magnetic Weyl semimetal (MWS) state and negative flat band magnetism in $Co_3Sn_2S_2$[13,14,17], and the topological Chern magnet in the quantum limit in $TbMn_6Sn_6$[12]. Defect excitations at atomic vacancies and adatoms, which are known to provide deeper understanding and reveal new physical properties of correlation topological materials[22–26], have yet to be explored in these kagome lattice materials.

Here, we study atomic defect excitations in the $Co_3Sn_2S_2$ by spin-polarized scanning tunneling microscopy (STM). Very recently, $Co_3Sn_2S_2$ has been discovered to exhibit novel phenomena such as surface-termination-dependent topological Fermi arcs[13] and disorder-induced elevation of intrinsic anomalous Hall conductance[18], making it an ideal platform to study the defect excitations and its correlation to the topological properties of the Weyl semimetal. Our main finding is the localized magnetic polarons nucleated around single atomic S vacancies on S-terminated surface in $Co_3Sn_2S_2$ by spin-polarized STM. They emerge as bound states in the conductance map with a three-fold rotation symmetry. Applying external magnetic fields up to ±6 T normal to the surface reveals that the binding energy

of the localized magnetic polaron linearly increases as a function of the field magnitude regardless of the field direction. This anomalous Zeeman response of a magnetic bound state has not been observed before and indicates dominant orbital magnetization contribution to the local magnetic moment (~1.35 $\mu_B$). Appreciable magneto-elastic coupling is also detected near the S-vacancy. We term this new excitation as a localized spin-orbit polaron (SOP) and argue that the large orbital magnetization has a topological origin associated with the Berry phase and the persistent circulating current due to the magnetoelectric effect of the topological magnet.

## Results

$Co_3Sn_2S_2$ has a layered structure that consists of two hexagonal planes of S and Sn as well as a $Co_3Sn$ kagome layer sandwiched between the S atoms (Fig. 1a). The magnetic Weyl semimetal has a Curie temperature of 177 K and a low-temperature out of the plane magnetic moment of about 0.3 $\mu_B$/Co[6,27,28]. We employed spin-polarized STM/S to study the cleaved surface of $Co_3Sn_2S_2$ (Fig. 1b). In STM topographic images, two types of cleaved surfaces were observed. Type-I surfaces show a hexagonal-like lattice with randomly distributed vacancies (Fig. 1c), while Type-II surfaces show a similar hexagonal-like lattice with adatoms and clusters (Fig. 1e).

**Identification of S-terminated surface of $Co_3Sn_2S_2$.** Weak bonds between S and Sn atoms offer a cleave plane and possibly lead to S-terminated and Sn-terminated surfaces. It is crucial to distinguish the two types of surfaces of the cleaved sample, which has been a controversial issue in previous studies[13,17,29,30]. In order to further identify the two surfaces, we conducted the local contact potential difference (LCPD) measurement on both surfaces using the low-temperature (4.5 K) nc-AFM/STM, which is based on a qPlus sensor[31]. Typical STM images of the Type-I and Type-II surfaces of $Co_3Sn_2S_2$ show the hexagonal lattice, with distinctive features of randomly distributed vacancies for Type-I

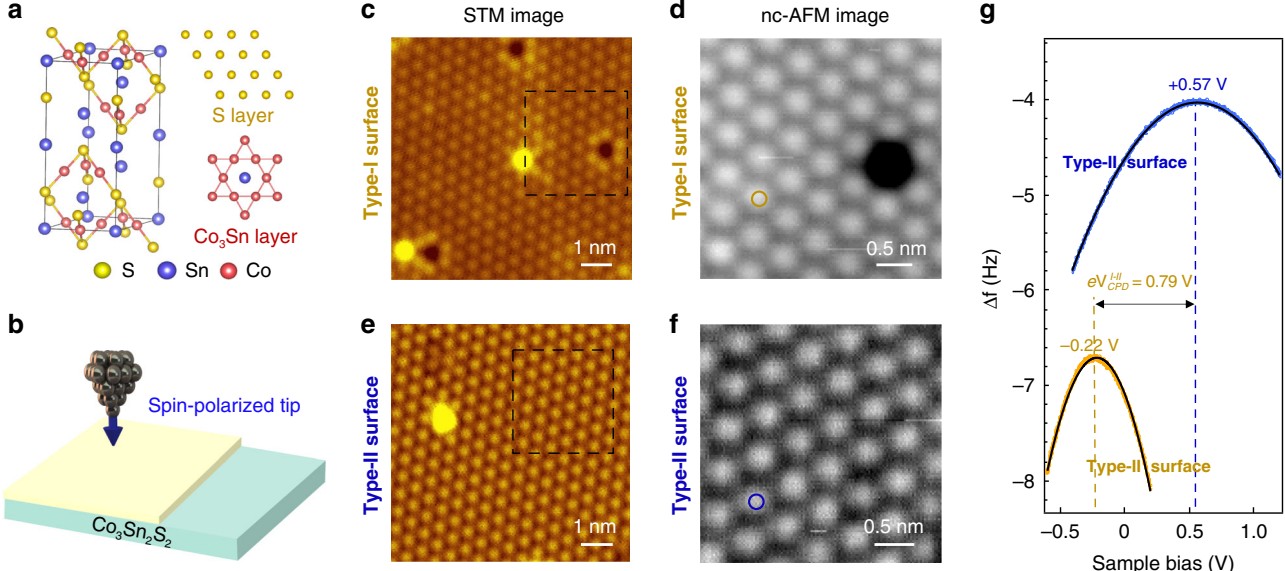

**Fig. 1 Identification of S-surface of $Co_3Sn_2S_2$ using nc-AFM/STM. a** Atomic structure of $Co_3Sn_2S_2$. **b** Schematic of spin-polarized STM measurements. **c** and **e** STM images on Type-I and Type-II surfaces, respectively. **d** and **f** Zoomed-in nc-AFM images of (**a**) and (**c**) marked by black squares, respectively. **g** $\Delta f(V)$ curves on the orange and blue circles in **d** and **f**. The maxima of the $\Delta f(V)$ parabolas are located at −0.22 V and +0.57 V for Type-I and Type-II surfaces, respectively. The work function difference between the two surfaces is 0.79 eV (refer to Supplementary Fig. 1 for more information). (STM scanning parameters: bias: $V_s = -400$ mV, setpoint: $I_t = 10$ pA); AFM scanning parameters: amplitude = 100 pm, frequency shift $\Delta f = -10$ Hz for **d** and −40 Hz for **f**).

surface and bright dots for Type-II surface. Zoomed-in nc-AFM images (Fig. 1d, f) of the two surfaces show consistent topography with the corresponding STM images. We then investigated the LCPD on the two surfaces by measuring frequency shift ($\Delta f$) with respect to the sample bias $V$ (Fig. 1g), respectively[32]. On the Type-I surface, the maximum of the parabola in the obtained $\Delta f$ ($V$) is $V^{\mathrm{I}}_{\mathrm{CPD}} = -0.22$ V, and on the Type-II surface, $V^{\mathrm{II}}_{\mathrm{CPD}} = +0.57$ V, which indicate that the work function of Type-I surface is ~0.79 eV higher than that of Type-II surface[33] (Supplementary Fig. 1). In addition, we calculated the work functions of the S (5.29 eV) and Sn (4.34 eV) terminated surfaces, respectively. The calculated work function value on the S-terminated surface is 0.95 eV higher than that on the Sn-terminated surface. This clearly demonstrates that the surface with the higher work function (Type-I surface) is the S-terminated surface.

**Localized excitations around S vacancies.** We next studied the properties of localized excitations by focusing on a region with S vacancies on the S-terminated surface. A large-scale topographic image shows randomly distributed S vacancies of the focused region (Fig. 2a) and a zoomed-in image depicts a single S-vacancy (Fig. 2c). A typical off-vacancy d$I$/d$V$ spectrum (black curve in Fig. 2b) closely resembles the one taken in a region free of S vacancies on the S-surface, exhibiting an energy range of suppressed and flat density of states of about 300 meV[13,29], and a broad hump around +50 meV, which originates from the topological surface states of the $Co_3Sn_2S_2$ magnetic Weyl semimetal[13,29]. There is also a peak sitting at the edge of the valence band at −350 meV. The calculated density of states (DOS) projected onto the S surface reproduces these features in the d$I$/d$V$ spectrum (Supplementary Fig. 2). Differently, the d$I$/d$V$ spectrum taken at the S-vacancy (orange curve in Fig. 2b) shows suppressed density of states around +50 meV and −350 meV. Meanwhile, a series of approximately equal-spaced spectral peaks emerge just above the valence band inside the region of suppressed density of states, indicating bound states formation at the S-vacancy. From statistical analysis, we determine the average energy spacing between the spectral peaks to be ~16 meV (Supplementary Fig. 3). Spatial distributions of these spectroscopic features were recorded in d$I$/d$V$ maps (Fig. 2d–h and Supplementary Fig. 4). The conductance map at +50 mV (Fig. 2d) shows an atomically modulated scattering pattern of the low-energy surface states at the S-vacancy. The map acquired at −350 mV (Fig. 2e) reveals a three-fold structure with suppressed intensity around the S-vacancy and all its six neighboring

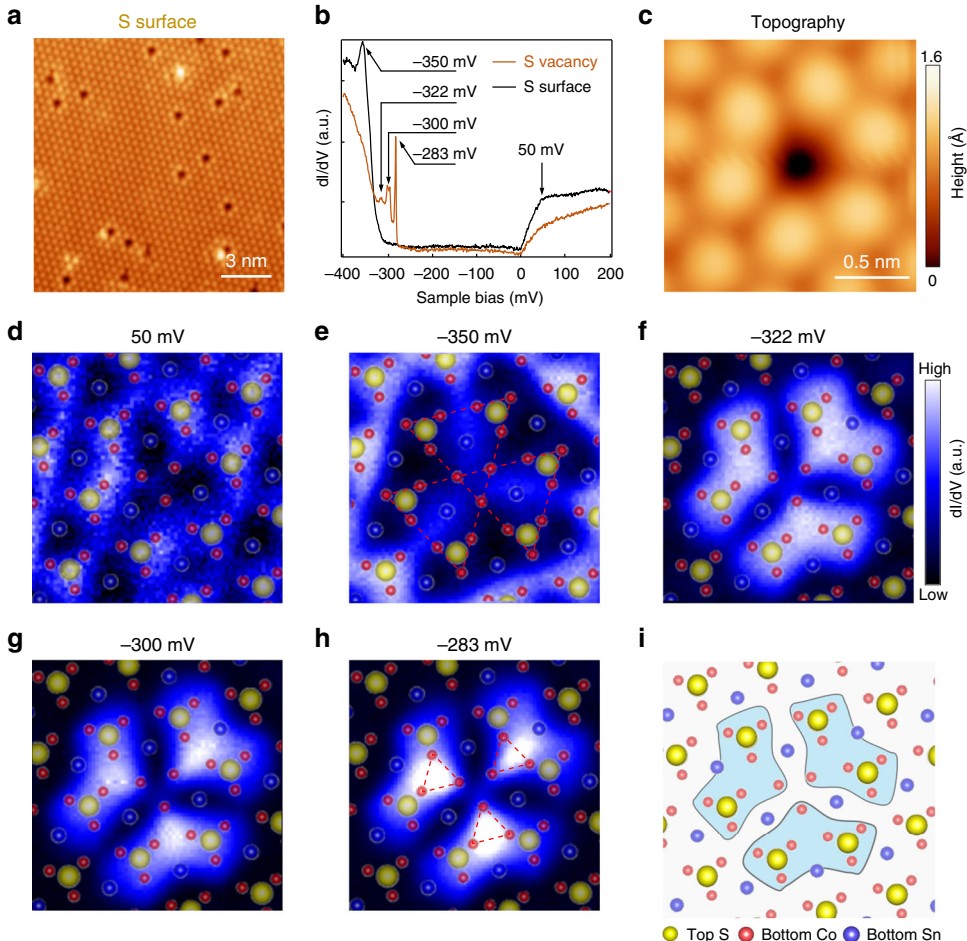

**Fig. 2 Localized excitations around a single S-vacancy at the S-terminated surface of $Co_3Sn_2S_2$. a** Atomic-resolution STM image of the S-terminated surface, showing randomly distributed single vacancies (scanning setting: bias: $V_s = -400$ mV, setpoint $I_t = 100$ pA). **b** d$I$/d$V$ spectra at (orange curve) and off (black curve) a single S-vacancy ($V_s = -400$ mV, $I_t = 500$ pA, $V_{\mathrm{mod}} = 0.5$ mV). **c** STM image of a S-vacancy ($V_s = -400$ mV, $I_t = 500$ pA). **d–h** d$I$/d$V$ maps of (**c**) at different energies: 50 meV (**d**), −350 meV (**e**), −322 meV (**f**), −300 meV (**g**), and −283 meV (**h**), respectively. Atomic structure of $Co_3Sn_2S_2$ is overlaid on each map, showing the correlation between the atomic structure and pattern in d$I$/d$V$ map ($V_s = -400$ mV, $I_t = 500$ pA, $V_{\mathrm{mod}} = 0.5$ mV). **i** Correlation between the atomic structure and the pattern in the d$I$/d$V$ map in **h**, showing that the spatial distribution of bound states is correlated to the underlying Co atoms. a.u., arbitrary units.

S atoms. Remarkably, the conductance maps (Fig. 2f–h) recorded at those three discernable d$I$/d$V$ peaks, i.e., at −322, −300, and −283 mV, all show localized flower-petal shaped patterns. Each pattern exhibits a three-fold rotation symmetry centered at the S-vacancy and the boundary of the pattern reaches as far as the six neighboring S atoms, which approximately defines the size of the bound states (Fig. 2i).

The peak residing at −283 mV (Fig. 2b) is the sharpest and the most localized one, which is referred to the primary bound state. It consists of two sub-peaks with small energy splitting (Supplementary Fig. 5). The brightest portion of its intensity is distributed over the three up-triangles closest to the vacancy, one in each of the three petals, in the underlying kagome lattice (highlighted by the red up-triangles in Fig. 2h), clearly demonstrating the driving force for the formation of the bound state polaron is the localization of the Co $d$-electrons by the vacancy potential of the missing S$^{2-}$ ion, which hybridize with the S $p$-electrons. Interestingly, the conductance map of the −283 mV state (Fig. 2h), also those of the −322 and −300 mV states (Fig. 2f, g), show a dramatic contrast/intensity reversal in comparison with the map of the −350 mV state (Fig. 2e), suggesting the −350 mV state be an antibound state, an affirmation of the bound state nature of the S-vacancy-induced polaron. A ~67 meV energy separation between the bound and antibound states allows us to estimate the "bandwidth" of the polaronic state.

**Magnetic properties of the bound state around S vacancies.** To investigate the magnetic properties of the bound states, we used spin-polarized Ni tips to measure d$I$/d$V$ spectra on the single S-vacancy (Fig. 1b). The spin-polarization of the Ni tip was calibrated on Co/Cu(111) (Supplementary Fig. 6). An external magnetic field of +0.6 T (−0.6 T) was applied to magnetize the Ni tip to be in the spin-up (spin-down) state. This tip was then used to measure the spin-dependent d$I$/d$V$ spectra at 0 T. We did not observe appreciable spin-polarized contrast in any defect-free regions of the S-terminated surface, indicating that the intact S-terminated surface is nearly non-magnetic (Fig. 3a and Supplementary Fig. 7). At the S-vacancy, the d$I$/d$V$ spectra, however, show strong magnetic contrast from approximately −350 mV to −280 mV where the bound states reside, indicating that the bound states are magnetic with a spin-down majority (Fig. 3b).

Figure 3c shows the spin-flip operation of the Ni tip[34,35], where we zoomed in to more closely show the primary bound state at ~−283 mV (marked by the black arrow in Fig. 3b). Particularly, a

spin-down tip was initially prepared, which gave a pronounced peak around −283 mV (left panel in Fig. 3c). The polarization of the tip was then flipped to spin-up by an external magnetic field of +0.6 T. Given this tip, the intensity of the −283 mV peak reduces while the peak position keeps unchanged (middle panel in Fig. 3c). After flipping the tip spin back to spin-down, the peak intensity restores (right panel in Fig. 3c). This result demonstrates that the bound states are magnetic polarons introduced by the S vacancies (Supplementary Fig. 8).

**Magnetic field response of the localized magnetic polaron.** We further investigated the nature of the localized magnetic polaron by measuring the magnetic field response of the spectral peaks using a normal W tip. The field is applied perpendicularly to the sample surface, ranging from −6 T to 6 T. The energies of the both sub-peaks of the primary bound state shift linearly toward the higher energy side ($\Delta E > 0$), independent of the direction of the magnetic field (Fig. 4a, b). This unusual behavior is reproducible on various individual S vacancies (Supplementary Fig. 9). The Zeeman effect $\Delta E = -\boldsymbol{\mu} \cdot \mathbf{H}$ indicates that the energy of a polarized (nondegenerate) magnetic state would decrease if the magnetic moment ($\boldsymbol{\mu}$) was parallel to the applied field but would increase if it was oriented antiparallel. The observed anomalous Zeeman response has two important physical implications. A $\Delta E$ independence of field directions implies that the magnetic field couples to the magnetization of the bound state, which simultaneously flips upon reversing the field direction. In addition, $\Delta E > 0$ indicates that the net magnetization is always oriented antiparallel to the direction of the magnetic field, indicative of a dominant diamagnetic orbital current contribution, in addition to the pseudospin (electron spin + atomic orbital) contribution in the presence of spin–orbit coupling.

To emphasize its orbital magnetic moment, we dub the magnetic bound state as a localized spin-orbit polaron (SOP). Fitting the two sub-peak positions as a linear function of the magnetic field (Fig. 4c), we obtained a slope of 75 μeV/T = 1.35 μ$_B$ for the effective magnetic moment of the SOP. The anomalous Zeeman response was recently observed for the flat portion of the itinerant band states corresponding to a peak in the density of states at low energies and attributed to the Berry phase induced orbital magnetization[17]. A crucial difference of the previous work from the present lies in that the energy shift confined to the flat part of the band states does not correspond to

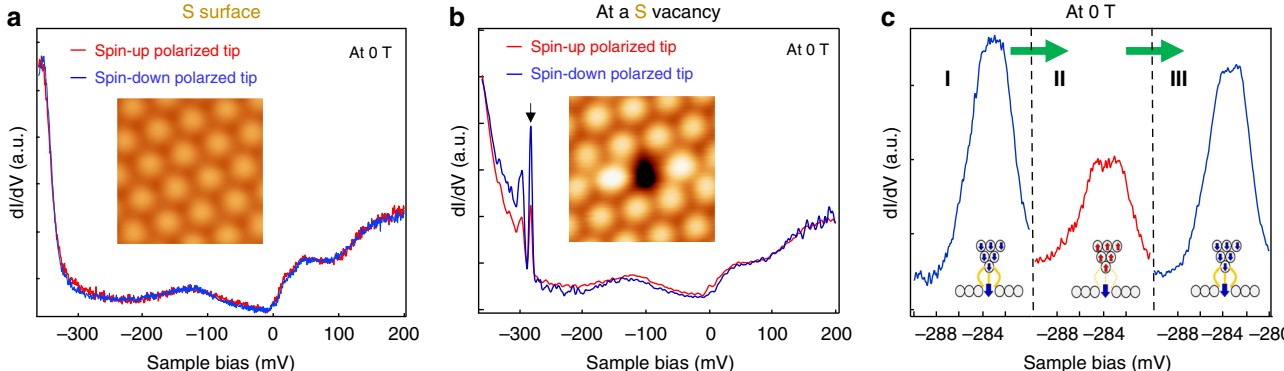

**Fig. 3 Spin-polarized bound states at a single S-vacancy. a** Spin-polarized d$I$/d$V$ spectra at the vacancy-free region on S-terminated surface using up-polarized tip (red curve) and down-polarized tip (blue curve), showing nearly no polarization contrast on pristine S-termined surface ($V_s = -400$ mV, $I_t = 100$ pA, Modulation $V_{mod} = 0.5$ mV). The inset shows the STM image (2 nm × 2 nm) of pristine S-terminated surface. **b** d$I$/d$V$ spectra at the center of a single S-vacancy, showing a spin-down majority behavior. The inset shows the STM image (2 nm × 2 nm) of the single S-vacancy ($V_s = -400$ mV, $I_t = 200$ pA, $V_{mod} = 0.5$ mV). **c** Spin-flip operation of the STM tip and the reproducible spectra at the S vacancy site. Left curve (curve I) corresponds to the initial spin-down tip polarization, the middle one (curve II) corresponds to spin-up tip polarization induced by a magnetic field, and the right one (curve III) corresponds to flipping the spin of the tip back to the initial spin-down polarization. ($V_s = -400$ mV, $I_t = 500$ pA, $V_{mod} = 0.5$ mV).

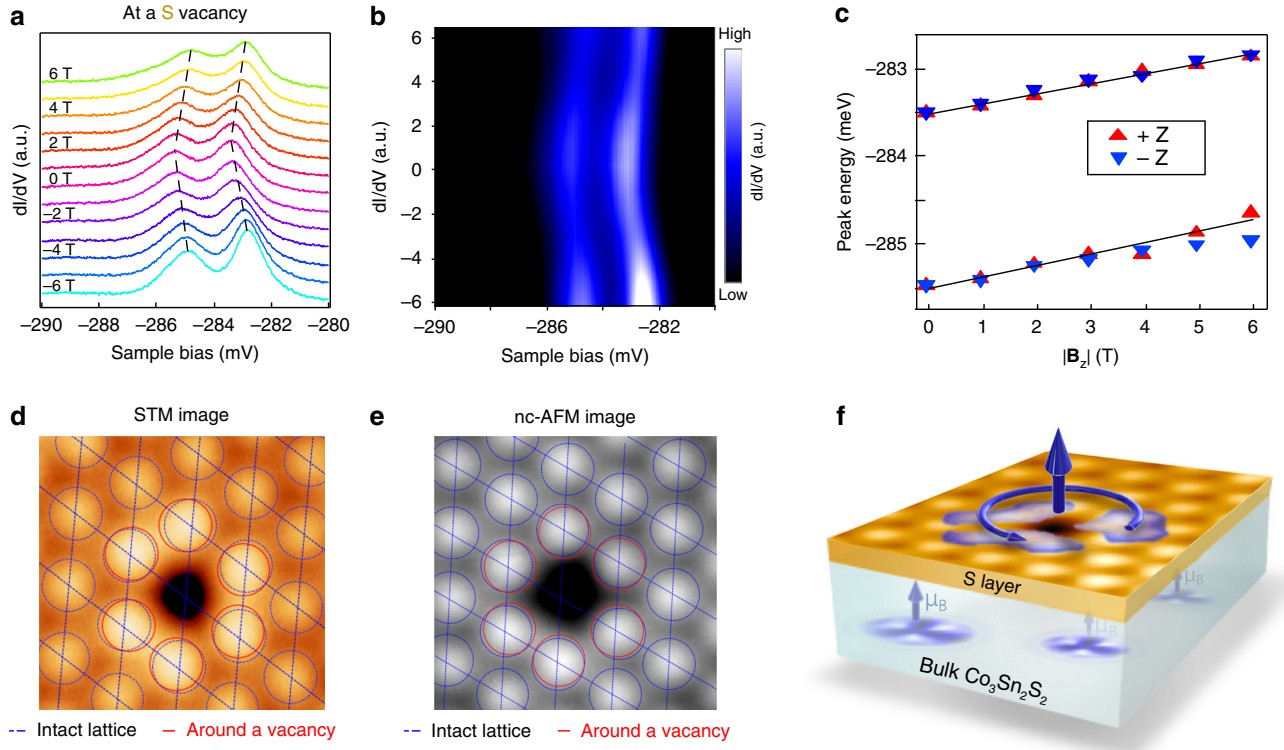

**Fig. 4 Anomalous Zeeman shift of the bound states and spin-orbit polaron at a single S-vacancy. a** $dI/dV$ spectra of the bound states in a magnetic field perpendicular to the sample surface from −6 T to 6 T, showing an approximately linear shift independent of the magnetic field direction. ($V_s =$ −400 mV, $I_t = 500$ pA, $V_{mod} = 0.5$ mV). **b** Intensity plot of **a**. **c** Energy shift of peak positions plotted as a function of the absolute value of magnetic field. **d**, **e** STM and nc-AFM images (2.3 nm × 2.3 nm) of the same single S-vacancy, showing appreciable lattice distortion (detailed analysis please refer to Supplementary Fig. 10) around the single S-vacancy (STM: $V_s = −400$ mV, $I_t = 500$ pA; AFM: $\Delta f = −6$ Hz). **f** Schematic illustration of the localized spin-orbit polaron in $Co_3Sn_2S_2$.

a net magnetic moment and the orbital magnetization averages out to approximately zero when all momentum states are considered[22]. The anomalous Zeeman shift observed here, however, comes from a localized bound state and corresponds to the overall magnetic field response of a net physical magnetic moment. The large orbital magnetization of the SOP most likely originates from the persistent circulating current around the S-vacancy due to the Berry phase and magnetoelectric effect of the topological Weyl semimetal[36].

The possible origin of the orbital magnetization of the SOP is new and intriguing. In dilute magnetic semiconductors (DMS)[37,38] such as (Ga, Mn)As, the doped magnetic ion $Mn^{2+}$ induces a magnetic polaron through the exchange coupling to the weakly correlated itinerant $p$-orbital electrons in the hole-band with spin–orbit coupling. While direct measurements of the magnetic properties of an isolated $Mn^{2+}$ are not available, the orbital contribution was estimated theoretically to dominate over the spin contribution to the hole magnetization[39], which is nevertheless too small (~5%) compared to the saturated ferromagnetic (FM) moment. The orbital magnetic moment was thus not crucial for DMS, although the physics of bound magnetic polarons[40–42] plays an important role in producing the FM order.

The physics in the topological Weyl semimetal $Co_3Sn_2S_2$ is different and much richer. First of all, the transition metal Co dominates both the itinerant and localized magnetism. The itinerant electrons come from the correlated Co $3d$ orbitals mixed with the Sn and S $p$-orbitals on the kagome lattice with frustrated kinetic energy[43]. The flat bands and the strong spin–orbit coupling endow the conduction electrons with robust spin-polarization and Berry phase. The "dopant" here is a $S^{2−}$ vacancy residing out of the

Co kagome lattice plane that causes the localization of the magnetic Co $3d$ electrons, forming a bound state SOP involving a cluster of atoms (Fig. 2i). It is known theoretically that the Berry phase and orbital magnetization can be established on electronic states localized on atomic clusters. Thus, the spin-polarized $d$-electrons naturally contribute to the orbital magnetization of the localized SOP, while the weakly correlated $p$-electrons and the $p$-$d$ exchange and charge transfer interactions play a minor role.

Second, more intriguingly, the topological properties of the Weyl semimetal can lead to the large orbital magnetization of the SOP through the magnetoelectric effect of the Weyl fermions. In a 3D strong topological insulator (TI), the Dirac fermion topological surface states, exchange-coupled to a magnetic impurity, produce persistent loop currents with large orbital magnetization around the local moment due to the magneto-electric effect[44]. Moreover, in 2D TI quantum dots of nanometer sizes, the circulating spin edge currents can be turned into charge current by a time-reversal symmetry-breaking magnetic field and produce large orbital magnetization[45]. In magnetic Weyl semimetals, the time-reversal symmetry is intrinsically broken and the surface states carry a well-defined chirality that is robust against disorder and localization[46]. Propagating chiral edge modes have recently been observed at step-edges on the surface of $Co_3Sn_2S_2$[30]. Thus, our experimental findings are consistent with the picture that the spin-polarized localized states generate persistent circulating diamagnetic currents and large orbital magnetization around the S-vacancy, which acts as a magnetic (anti-)dot extending to the neighboring S atoms. Most recently, such circulating persistent currents induced by the topological surface states with diamagnetic moment around quantum dots in

Weyl semimetals were theoretically predicted[36], which provides further and more direct support for our experimental findings and theoretical interpretation of the S-vacancy as a quantum dot.

In this sense, the localized SOP in the magnetic Weyl semimetal has an inherent topological origin. While in-depth theoretical works are clearly necessary to account for the rich physics revealed by our experiments, some of the aspects can be captured qualitatively by density functional theory (DFT) calculations with spin–orbit coupling, where the total magnetic moment and its orbital content can be studied (Supplementary Fig. 8).

We further considered the magneto-elastic coupling within the localized SOP. The lattice distortion around the S vacancies measured by STM (Fig. 4d) and nc-AFM (Fig. 4e) shows appreciable local atomic displacements at 0 T. The measured average nearest atomic distances around the S-vacancy determine the local atomic displacement ratio as its percentage change from the average distance of an intact region. The displacement ratio significantly decreases with increasing strength of the field applied along the c-axis (Supplementary Fig. 10). Remarkably, one-third of the displacement ratio can be manipulated by a magnetic field up to 6 T. These observations further support the SOP nature of the bound states and the strong magneto-elastic coupling in $Co_3Sn_2S_2$.

## Discussion

The discovery of the localized SOP (Fig. 4f) opens a novel route for manipulating the magnetic order and the topological phenomena in Weyl semimetal $Co_3Sn_2S_2$. The STM observed S vacancies on S-terminated surfaces cleaved at low temperature (6 K) indicate the presence of bulk S vacancies, where localized SOPs are expected to nucleate with similar physical properties. DFT calculations reveal that the bulk vacancies are magnetic and significantly enhance the magnetic moment of neighboring Co atoms (Supplementary Fig. 11). The overall density of the SOP can thus be controlled by varying the S pressure and temperature during sample synthesis[47]. In light of this, the enhanced FM moment, experimentally observed with increasing S deficiency in the bulk[47], is most likely a consequence of the nucleation of the magnetic polarons at the S vacancies. A higher SOP density ramps up the scattering of the carriers by the spin-orbit exchange field as well as the exchange interaction between the SOPs, unleashing the potential for more robust time-reversal-symmetry-breaking topological phenomena such as the anomalous Hall and Nernst transport at higher temperatures. Given the role of magnetic dopants in dilute magnetic semiconductors[40–42,48], the vacancy-induced SOP may provide a new path toward generating large magnetic moments in correlated non-magnetic topological semimetals. Similar to the single defect engineering for scalable qubits and spin sensors in diamond[49], controlled engineering of the SOPs may pave the way toward practical applications in functional quantum devices.

## Methods

**Single crystal growth of $Co_3Sn_2S_2$.** The single crystals of $Co_3Sn_2S_2$ were grown by flux method with Sn/Pb mixed flux. The starting materials of Co (99.95% Alfa), Sn (99.999% Alfa), S (99.999% Alfa) and Pb (99.999% Alfa) were mixed in molar ratio of Co:S:Sn:Pb = 12:8:35:45. The mixture was placed in $Al_2O_3$ crucible sealed in a quartz tube. The quartz tube was slowly heated to 673 K over 6 h and kept over 6 h to avoid the heavy loss of sulfur. The quartz tube was further heated to 1323 K over 6 h and kept for 6 h. Then the melt was cooled down slowly to 973 K over 70 h. At 973 K, the flux was removed by rapid decanting and subsequent spinning in a centrifuge. The hexagonal-plate single crystals with diameters of 2–5 mm were obtained. The composition and phase structure of the crystals was checked by energy-dispersive X-ray spectroscopy and X-ray diffraction, respectively.

**Scanning tunneling microscopy/spectroscopy.** The samples used in the experiments cleaved in situ at 6 K and immediately transferred to an STM chamber. Experiments were performed in an ultrahigh vacuum ($1 \times 10^{-10}$ mbar) ultra-low temperature STM system (40 mK) equipped with 9-2-2 T magnetic field. All the scanning parameter (setpoint voltage and current) of the STM topographic images are listed in the captions of the figures. Unless otherwise noted, the differential conductance (d$I$/d$V$) spectra were acquired by a standard lock-in amplifier at a modulation frequency of 973.1 Hz. Non-magnetic tungsten tip was fabricated via electrochemical etching and calibrated on a clean Au(111) surface prepared by repeated cycles of sputtering with argon ions and annealing at 500 °C. Ferromagnetic Ni tip was applied in the spin-polarized STM measurement. The Ni tip was fabricated via electrochemical etching of Ni wire in a constant-current mode[50]. To calibrate the spin-polarization of Ni tip, the as-prepared Ni tip has been applied to resolve magnetic-state-dependent contrast of Co islands grown on a Cu(111) surface in spin-polarized STM experiments (details are shown in Supplementary Fig. 6). External magnetic fields of +0.6 T (−0.6 T) were applied to magnetize the Ni tip to be in the spin-up (spin-down) states. This tip was then used to measure the spin-dependent d$I$/d$V$ spectra at 0 T on the same atomic location. We also reproduce the spin-polarized measurements result by applying +0.2 T (−0.2 T) external magnetic fields.

**Non-contact atomic force microscopy (nc-AFM).** nc-AFM measurements were performed at LHe temperature with the base pressure lower than $2 \times 10^{-10}$ mbar. All nc-AFM measurements were performed using a commercial qPlus tuning fork sensor in frequency modulation mode with a Pt/Ir tip, while all STM topographic images were simultaneously acquired in constant-current mode. The resonance frequency is about 27.9 kHz, and stiffness about 1800 N/m. The imaging heights for all nc-AFM measurements are reported in figure captions referred to the STM tunneling junction height on clean Ag(100) substrate, which is −30 mV and 10 pA. All STM and nc-AFM images were processed using free and open source software Gwyddion.

**Density functional theory calculations.** Quantum mechanical calculations based on DFT were performed by using the Vienna Ab initio Simulation Package (VASP)[51,52]. The projector augmented wave (PAW)[53] method was employed, and the Perdew–Burke–Ernzerhof (PBE)[54] type of exchange correlation functional was used.

The slab models containing six Co-Sn layers and extra S or Sn layers were used to simulate the S-terminated and Sn-terminated surfaces. S surfaces with a single S-vacancy were simulated by a $4 \times 4$ supercell. In structural relaxation, the atoms in the two middle $Co_3Sn$ layers were fixed, while atoms in other layers were totally relaxed. The vacuum layers of the slab models were larger than 15 Å. A S vacancy in bulk is modeled in a $2 \times 2 \times 2$ supercell.

The wavefunctions were expanded in plane waves with a kinetic energy cutoff of 400 eV. For pristine S-terminated and Sn-terminated surfaces of $Co_3Sn_2S_2$, the k-points sampling was $8 \times 8 \times 1$, generated by Monkhorst-Pack grids with the origin at the Γ-point. The structures were relaxed until the energy and residual force on each atom were smaller than $10^{-6}$ eV and 0.001 eV/Å, respectively. For S-terminated surface with a single vacancy, the k-points sampling was with only the Γ-point. For the S vacancy in the bulk, the k-points sampling was $3 \times 3 \times 3$, generated by Monkhorst-Pack grids with the origin at the Γ-point. The structures were relaxed until the energy and residual force on each atom were smaller than $10^{-4}$ eV and 0.01 eV/Å, respectively. Non-collinear calculations considering the spin–orbit coupling (SOC) were carried out to determine the magnetic moment direction of each layers. The slab model calculations show that the easy axis is in the z (out of plane) direction with the vacancy magnetic moment parallel to that of Co atoms.

With these parameters, the optimized lattice constants of bulk $Co_3Sn_2S_2$ are 5.37 Å and 13.15 Å along a and c directions, respectively. The magnetic moment is 0.35 $\mu_B$, −0.02 $\mu_B$, −0.03 $\mu_B$, and 0 $\mu_B$ for Co, Sn in $Co_3Sn$ plan, Sn in Sn layer, and S, respectively. The magnetic moment and projected density of states of the polaron were calculated by projecting the wave function onto an empty sphere sitting at the S vacancy site. The radius of this sphere is chosen as 1.164 Å, which is the Wigner–Seitz radius of S.

## Data availability

Data measured or analyzed during this study are available from the corresponding author on reasonable request.

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

## Acknowledgements

We thank Qing-Feng Sun, Fu-Chun Zhang, Gang Su, and Werner Hofer for helpful discussions. The work is supported by grants from the National Natural Science Foundation of China (61888102, 11974422 and 11974394), the National Key Research and Development Projects of China (2016YFA0202300, 2017YFA0206303, 2018YFA0305800, 2019YFA0308500 and 2019YFA0704900), and the Chinese Academy of Sciences (XDB28000000, 112111KYSB20160061). Z.Q.W. is supported by the US DOE, Basic Energy Sciences Grant No. DE-FG02-99ER45747.

## Author contributions

H.-J.G. designed the experiments. Y.X., H.C., B.H., G.L., P.F., G.Q., L.C., and C.S. performed STM experiments with guidance of H.-J.G. L.H., Q.Z., R.M., and L.X. performed AFM/STM experiments. E.L., J.S., Q.W., Q.Y., H.L., H.Y., W.W., and B.S. prepared samples. Z.W. proposed the model and Y.G., X.Z., Y.Y.Z., S.D., and W.J. carried out theoretical calculations. All of the authors participated in analyzing experimental data, plotting figures, and writing the manuscript. H.-J. G. and Z.W. supervised the project.

## Competing interests

The authors declare no competing interests.
