## [Peer Review File · Nature Communications]

This manuscript has been previously reviewed at another journal that is not operating a transparent peer review scheme. This document only contains reviewer comments and rebuttal letters for versions considered at *Nature Communications*.

Reviewers' Comments:

Reviewer #2:

Remarks to the Author:

Authors resolved most of my concerns. If the points below are satisfactorily addressed, I will consider recommending the paper for publication in *Nature Communications*.

1. "Response 3-d: We inserted the following sentences to pointing out the direction of the calculated vacancy magnetic moment in the Method part as follows: Non-collinear calculations considering the spin-orbit coupling (SOC) were carried out to determine the magnetic moment direction of each layers. It is found that for the slab model calculations, the easy axis is in the z (out of plane) direction."

I appreciate this clarification, but what I meant is the direction of the vacancy magnetic moment with respect to that of Co. are they parallel or antiparallel?

2. "Response 4: In the plot of the folded bands (now ED_Fig. 7a), there is a gap but with a width of ~ 0.1 eV. In the DOS plot, because of the 0.05 eV smearing width we used, the gap appears as a dip feature, which does exist in ED_Fig. 6c."

I suggest that the authors write this explicitly, otherwise a reader might be confused exactly as I did: a gap in one set of data, while no gap in the other for exactly the same thing – the electronic spectrum of the same system.

3. "Response 8: We thank the reviewer for raising this question. ED_Fig. 1 was originally used to help identify the S surface determination using only STM. Now, since we have new AFM data to directly identify the S surface, ED_Fig. 1 and its corresponding discussions are not needed and have been removed in this revision. In light of this, the discussion of this issue is no-long necessary."

This is still unsatisfactory. In all theoretical data the bound state is located in the conduction band, while in the experiment it is occupied. No reasonable explanation for that is given. Removing data from the extending data does not resolve this problem.

4. Finally, the authors should elaborate more the introduction part. Basically, the intro of this paper is only one paragraph. It is too general, the statements are hazy, the setting in of the problem that is going to be solved/addressed is not clear. No overview of relevant literature, description of the state-of-the-art. All this should be improved keeping in mind that *Nature Communications* is multidisciplinary journal

Point-by-point response to the comments from the reviewers

We thank the reviewer 2[#] for his/her time and effort for reviewing our manuscript. We have addressed all the criticisms and comments point-by-point below and revised the manuscript accordingly. In this response letter, comments from the reviewers are summarized in black italic typeface. Our responses are in regular blue typeface. Our changes to the text are in red.

Response to Reviewer 2[#]

Overall Critical Comment:

“Authors resolved most of my concerns. If the points below are satisfactorily addressed, I will consider recommending the paper for publication in Nature Communications.”

Response: We thank the reviewer for the constructive comments and suggestions of our work. In the following, according to reviewer’s comments and suggestions, we have done further revisions to address all of the concerns from the reviewer, and made the revisions correspondingly.

Comment 1. *“Response 3-d: We inserted the following sentences to pointing out the direction of the calculated vacancy magnetic moment in the Method part as follows: Non-collinear calculations considering the spin-orbit coupling (SOC) were carried out to determine the magnetic moment direction of each layers. It is found that for the slab model calculations, the easy axis is in the z (out of plane) direction.”*

I appreciate this clarification, but what I meant is the direction of the vacancy magnetic moment with respect to that of Co. are they parallel or antiparallel?

Response 1: The direction of the vacancy magnetic moment is parallel to that of Co. We inserted the following sentences in the Method part.

“the easy axis is in the z (out of plane) direction with the vacancy magnetic moment parallel to that of the Co atoms.”

Comment 2. *“Response 4: In the plot of the folded bands (now ED_Fig. 7a), there is a gap but with a width of ~ 0.1 eV. In the DOS plot, because of the 0.05 eV smearing width we used, the gap appears as a dip feature, which does exist in ED_Fig. 6c.”*

I suggest that the authors write this explicitly, otherwise a reader might be confused exactly as I did: a gap in one set of data, while no gap in the other for exactly the same thing - the electronic spectrum of the same system.

Response 2: We took the reviewer's suggestions and inserted a description right after Extended Data Fig. 8a and 8b: "This band gap is reflected as a dip feature in Extended Data Fig. 7c, which is due to the 0.05 eV smearing width used in plotting the figure."

Comment 3. "Response 8: We thank the reviewer for raising this question. ED_Fig. 1 was originally used to help identify the S surface determination using only STM. Now, since we have new AFM data to directly identify the S surface, ED_Fig. 1 and its corresponding discussions are not needed and have been removed in this revision. In light of this, the discussion of this issue is no-long necessary. "

This is still unsatisfactory. In all theoretical data the bound state is located in the conduction band, while in the experiment it is occupied. No reasonable explanation for that is given. Removing data from the extending data does not resolve this problem.

Response 3: We thank the reviewer for mentioning this point again. The reviewer's concerns lies in that the localized bound state sitting above the Fermi level and is thus unoccupied, which appears different than the experimental observation. Following reviewer's suggestions, we have examined the role of substrate terminations of the bottom layer, considered the interaction effects using the modified Becke-Johnson form and accounted for the strong Coulomb interaction around the bound state by adding an appreciable U value, i.e. 1 to 4 eV with a step of 1 eV, to the Co atoms around the S vacancy. All these calculations show that the bound state residing above the Fermi level, which implies that such inconsistency most likely lies beyond the DFT based calculations.

We would like to reiterate that the overall line shape and features of the DFT calculations agrees reasonably with the experiment, provided an overall shift in the Fermi energy of about 0.35 eV. Moreover, this discrepancy is not specific to the vacancy bound state calculation. The agreement between the measured density of states away from vacancies and the DFT calculations in the absence of vacancies also requires a similar overall shift, which is consistent with the findings from other groups [A similar offset is also required to match the dI/dV data. Page 2 in Supplementary Information of Morali, N. *et al. Science* **365**, 1286–1291 (2019).]

The required offset of the Fermi energy in comparisons between DFT calculations and spectroscopic measurements on the surface is not uncommon, and is usually attributed

to carrier induced chemical potential shift and/or correlation effects beyond the DFT framework. As we discussed in the manuscript, our samples, most likely, contain an appreciable amount of S vacancies, leading to electron doping of the sample. Transport measurements confirmed the existence of electron carriers [please refer to: Fig. 1f of Liu, E. *et al. Nat. Phys.* **14**, 1125–1131 (2018)], which usually shifts the calculated LDOS downwards when comparing with the measured dI/dV spectrum.

After careful double check with our DFT results and experimental details, the vacancy induced electron doping effect appears the most likely reason for the discrepancy. We agree that this is an open issue to be addressed in the future. Doping effect or correlation effect usually leads to similar discrepancy between calculated LDOS and the measured dI/dV spectrum.

We restored previous ED_Fig. 1 (now Supplementary_Fig. 2), revised captions of SI_Fig. 2 and SI_Fig. 8 as following, and we will try to address this issue in future experiments and theoretical calculations.

In SI_Fig. 2 caption:

“It is worthy to mention that required offset of the Fermi energy in comparisons between DFT calculations and spectroscopic measurements on the surface is usually attributed to charge carrier induced chemical potential shift and/or correlation effects. A similar offset is also required to match the dI/dV data in previous studies of $\text{Co}_3\text{Sn}_2\text{S}_2$ ¹³.”

In SI_Fig. 8 caption:

“The overall line shape and features of the DFT calculations reasonably agrees with those of the experiment, with an overall shift of about 0.35 eV to the Fermi energy (the energy shift has been discussed in Supplementary Fig. 2b). It’s worthy to point out that there is a ~0.1 eV gap in the spin-down channel...”

Comment 4. *Finally, the authors should elaborate more the introduction part. Basically, the intro of this paper is only one paragraph. It is too general, the statements are hazy, the setting in of the problem that is going to be solved/addressed is not clear. No overview of relevant literature, description of the state-of-the-art. All this should be improved keeping in mind that Nature Communications is multidisciplinary journal.*

Response 4: We agree with the reviewer on this comment. We have rewritten the introduction part in the revised manuscript:

“The transition metal based kagome lattice compounds have merged recently as a novel materials platform for unveiling and exploring the rich and unusual physics of geometric frustration, correlation and magnetism, and the topological behaviors of the

quantum electronic states^{1–18}. These are layered crystalline materials where the transition metal elements occupy the vertices of the two-dimensional network of corner-sharing triangles, supporting electronic band structures with Dirac crossings and nearly flat bands with strong spin-orbit coupling^{19–21}. The prototype materials include Fe_3Sn_2 ^{1–3}, FeSn ⁴, $\text{Co}_3\text{Sn}_2\text{S}_2$ ^{5,6}, CoSn ⁷, Mn_3Sn ^{8–11}, and rare earth (Re) ReMn_6Sn_6 ^{12,15}. They exhibit different magnetic ground states, such as ferromagnetic (Fe_3Sn_2 , $\text{Co}_3\text{Sn}_2\text{S}_2$), antiferromagnetic (FeSn , Mn_3Sn), and paramagnetic (CoSn), and often anomalous transport properties of underlying topological origins^{5,6,16}. Remarkable phenomena have been reported recently, such as the giant spin-orbit tunability of the Dirac mass and electronic nematicity in Fe_3Sn_2 ^{1,2}, the magnetic Weyl semimetal (MWS) state and negative flat band magnetism in $\text{Co}_3\text{Sn}_2\text{S}_2$ ^{13,14,17}, and the topological Chern magnet in the quantum limit in TbMn_6Sn_6 ¹². Defect excitations at atomic vacancies and adatoms, which are known to provide deeper understanding and reveal new physical properties of correlation topological materials^{22–26}, have yet to be explored in these kagome lattice materials.

Here, we study atomic defect excitations in the $\text{Co}_3\text{Sn}_2\text{S}_2$ by spin-polarized STM. Very recently, $\text{Co}_3\text{Sn}_2\text{S}_2$ has been discovered to exhibit novel phenomena such as surface-termination dependent topological Fermi arcs¹³ and disorder-induced elevation of intrinsic anomalous Hall conductance¹⁸, making it an ideal platform to study the defect excitations and its correlation to the topological properties of the Weyl semimetal. Our main finding is ...”